# BRMI-Net: Deep Learning Features and Flower Pollination-Controlled Regula Falsi-Based Feature Selection Framework for Breast Cancer Recognition in Mammography Images

**DOI:** 10.3390/diagnostics13091618

**Published:** 2023-05-03

**Authors:** Shams ur Rehman, Muhamamd Attique Khan, Anum Masood, Nouf Abdullah Almujally, Jamel Baili, Majed Alhaisoni, Usman Tariq, Yu-Dong Zhang

**Affiliations:** 1Department of Computer Science, HITEC University, Taxila 47080, Pakistan; 2Department of Circulation and Medical Imaging, Faculty of Medicine and Health Sciences, Norwegian University of Science and Technology (NTNU), 7491 Trondheim, Norway; 3Department of Information Systems, College of Computer and Information Sciences, Princess Nourah bint Abdulrahman University, P.O. Box 84428, Riyadh 11671, Saudi Arabia; naalmujally@pnu.edu.sa; 4College of Computer Science, King Khalid University, Abha 61413, Saudi Arabia; jabaili@kku.edu.sa; 5College of Computer Science and Engineering, University of Ha’il, Ha’il 81451, Saudi Arabia; m.alhaisoni@uoh.edu.sa; 6Management Information System Department, College of Business Administration, Prince Sattam Bin Abdulaziz University, Al-Kharj 16278, Saudi Arabia; u.tariq@psau.edu.sa; 7Department of Informatics, University of Leicester, Leicester LE1 7RH, UK; yudong.zhang@leicester.ac.uk

**Keywords:** breast cancer, data augmentation, deep learning, features fusion, feature optimization

## Abstract

The early detection of breast cancer using mammogram images is critical for lowering women’s mortality rates and allowing for proper treatment. Deep learning techniques are commonly used for feature extraction and have demonstrated significant performance in the literature. However, these features do not perform well in several cases due to redundant and irrelevant information. We created a new framework for diagnosing breast cancer using entropy-controlled deep learning and flower pollination optimization from the mammogram images. In the proposed framework, a filter fusion-based method for contrast enhancement is developed. The pre-trained ResNet-50 model is then improved and trained using transfer learning on both the original and enhanced datasets. Deep features are extracted and combined into a single vector in the following phase using a serial technique known as serial mid-value features. The top features are then classified using neural networks and machine learning classifiers in the following stage. To accomplish this, a technique for flower pollination optimization with entropy control has been developed. The exercise used three publicly available datasets: CBIS-DDSM, INbreast, and MIAS. On these selected datasets, the proposed framework achieved 93.8, 99.5, and 99.8% accuracy, respectively. Compared to the current methods, the increase in accuracy and decrease in computational time are explained.

## 1. Introduction

Breast cancer is the most common disease among women, and it affects the breast region and spreads to other body parts. Breast cancer affects the breast tissue and is known to be the second most widely spread disease in the world [1]. In an investigation by the World Health Organization (WHO), 8.4% of breast cancer patients received a diagnosis, and 6.6% passed away [2]. Breast cancer affects more than 8% of women at some point in time [3]. Breast cancer claimed the lives of 43,250 women in 2022. Breast cancer-related tumors are classified as benign or malignant [4]. A malignant tumor spreads to other organs, whereas a benign tumor does not spread to the rest of the body [5]. There are numerous imaging tools available for early breast cancer treatment. A pathological biopsy is one of the most accurate methods; according to evidence-based medicine, patients with cancerous tumors should avoid having too many biopsies performed in order to stop cancer cell growth and metastasis. As a result, pathological examinations are frequently performed prior to chemotherapy [6]. Furthermore, computer-aided radiologists (CAD) discovered that diagnosis aids in the early detection of breast cancer in ultrasound images (US) while decreasing operator-dependent US imaging behavior [7].

Mammography monitoring is one of the most important methods for preventing breast cancer. An X-ray is used to examine the breast in this technique [8]. Asymptomatic women are subjected to mammography screenings and procedures in order to detect early, clinically undiagnosed breast cancer [9]. The best imaging method for detecting microcalcifications and ductal carcinoma in situ is mammography (DCIS) [10]. The widespread adoption of screening mammography is likely to blame for the rise in the overall incidence of DCIS during the past 20 years [11]. Prior to classification, previous research used a variety of techniques to develop computer-aided diagnosis (CAD) systems, including images. To improve the images’ ability to identify benign or malignant tissue, edge detection, wavelet transform, microcalcification detection, and region of interest (ROI) segmentation were used [12].

Causes such as noise, poor image quality, and unrelated parts will have an impact on the most common characteristics in the field of medical images [13]. Preprocessing techniques are used to address this issue [14]. The preprocessing technique is the technique that improves the image quality [15]. The next process is to extract features from the targeted region [16]. After passing these features to a classifier, it determines whether these mammogram images are normal or abnormal [17]. Feature selection is a procedure that follows feature extraction and is frequently used in machine learning [18]. The primary goal of feature selection is to remove redundant features from the original data [19]. The many characteristics of machine learning algorithms include effective performance on healthcare datasets involving images such as X-rays, and various blood samples [20].

Deep learning has recently significantly improved in several areas, including cell segmentation, skin melanoma, and hemorrhage detection [21]. Deep learning has been shown to be effective in medical imaging, particularly in COVID-19, breast cancer detection, the diagnosis of brain tumors, Alzheimer’s disease, and a variety of other conditions. A convolutional neural network (CNN) is a deep learning-based architecture with many layers. CNN converts pixels in an image into features [22]. Later, the characteristics are used to identify and classify infections. The features were extracted by CNN from the original images. The collection of irrelevant features extracted from the raw images also influences how well the classification performs. Choosing only the most relevant features is critical for improving classification precision. An active research area is the selection of the best features from the extracted initial features. Particle swarm optimization (PSO), genetic algorithm (GA) [23], and a few other selection algorithms are described and used in the literature for medical imaging [24]. These methods focus on the most beneficial subset of features rather than the entire feature field. The ability of feature selection methods to improve system accuracy while reducing computation time is their primary advantage. However, when selecting the best features, some critical features are sometimes overlooked, reducing the system’s precision. As a result, experts in computer vision developed feature fusion techniques. The system is more precise and has more predictors as a result of the fusion procedure. Two popular fusion techniques are serial-based fusion and parallel fusion.

Several computer vision-based methods for detecting and classifying breast cancer using mammography images have been presented in recent years [25]. Some worked on segmentation, while others focused on feature extraction and feature extraction from raw images [26]. In addition, researchers used and worked on the preprocessing step to enhance the affected area for better feature extraction and to improve the contrast and noisy images [27]. For example, Dilovan et al. [28] presented a method for deep learning in breast cancer. In this suggested approach, the region is selected using machine learning techniques of interest. The region of interest is separated and divided into different blocks. The researchers extracted deep features and optimized them by utilizing a genetic algorithm. As a result, they achieved 98.7% accuracy. The limitation of this proposed method was that there were multiple features from each diagnosed block. Shahan et al. [29] presented a method that utilizes deep learning for breast cancer. In the proposed methodology, they collected information from various sources, fused the datasets, and simulated them using breast image data. They obtained 99% accuracy as a result. The proposed method’s drawback was that computational intelligence techniques could be used to increase the system’s accuracy.

Huynh et al. [30] described a procedure for treating breast cancer using transfer learning from a deep convolution network. They used and extracted features from CNN to train the support vector machine according to this proposed approach. A 90% accuracy rate was achieved. The deep learning strategy used in the suggested method, which uses large datasets and potent computational resources, has a drawback. Ghada et al. [31] reported a deep learning-based method for classifying breast cancer cases. Retina net, a deep learning-based model, was used in this suggested classification technique. The results were 97% accurate as a result. The shortcomings of the proposed technique were data duplication, imbalance, and inconsistent object detectors. Jing et al. [32] reported a technique for identifying breast cancer using the deep learning Ada Boost algorithm. They used CNN LSTM and advanced computational methods in the suggested methodology. They had a 98.3% success rate. The suggested approach’s disadvantage was the required sizeable quantity of training data. Neslihan et al. [33] presented a deep-learning image classification method for treating breast cancer. In the suggested approach, they used a CNN-based model to forecast both the likelihood of malignant tumors and the degree of image malignancy. They were accurate to 82.13%. The task-wise early stopping in the multitasking design was the method’s flaw. Naresh et al. [34] presented the use of a deep learning system for detecting breast cancer. In the proposed approach, they compared deep learning algorithms to machine learning algorithms using a deep learning neural network model. Their success percentage was 99.67%. The lack of data for expansion was a problem with the suggested approach. A breast cancer treatment method reported by Yeman et al. [35] used deep learning. This technique used a patch-based multi-input CNN to detect breast tumors. As a result, they had a 92% accuracy rate. The proposed method’s disadvantage was that reliability and detection accuracy could be improved by training with a larger data set. Luqman et al. [36] presented a method for the semantic segmentation of image data for breast cancer using deep learning. They used RCNN and a transfer learning preprocessing algorithm in this approach. They were, therefore, 98% accurate. The suggested methodology’s limitations were fewer training instances, sampling disease, and class imbalance. Aruna et al. [37] described a technique for the early detection of breast cancer using statistical tools. Support vector machines and Naive Bayes were contrasted with other data mining methods. They discovered the best predictor using WEKA. They were entirely accurate. The SVMs, both linear and non-linear, could not be analyzed using the suggested approach. Kundan et al. [38] presented a convolutional neural network-based treatment for breast cancer. They used a CNN-based model and classified the features in the suggested methodology. An accuracy of 90% was attained. The proposed method’s limitation was the small size of the data collection. Bindu et al. [39] described a method for categorizing breast cancer using an artificial neural network with the ideal amount of inputs. They classified the features using the feed-forward algorithm in the suggested approach. They obtained 90% accuracy as a result. The method’s flaw was that determining the tumor variety took longer and cost more money. Umar et al. [40] presented a technique for detecting breast cancer that classified mammography images using a convolution neural network. They used the CNN-based model in this methodology for training, and applied the winner filter’s preprocessing procedure. They then divided the crucial data into segments (masses). They were, therefore, 98% accurate.

### 1.1. Major Challenges

The approaches discussed above emphasized preprocessing and deep learning-based techniques to increase classification accuracy. However, they had to deal with limited data, parallels between benign and malignant masses, and redundant feature data. The most crucial stage in any computer-aided diagnostic system is feature engineering, according to literature reviews. Additionally, it is noted that the experts did not consider the optimization process, which can improve the accuracy and shorten the computation time. The key challenges which we consider in this work are as follows: (i) It is difficult to manually separate malignant and benign extracted slices from original compact images. (ii) Because there are fewer malignant images than benign lesions, this affects the selected deep learning model during the training process. (iii) Regarding the small growth of lesions, the visual has a high degree of similarity between benign and malignant lesions, which may distort the correct classification. (iv) To improve accuracy, selecting important features ignores some essential features found by several optimization algorithms [41], due to their stopping condition and search criteria [42]. We can address these issues using a mammography image classification for breast cancer using a deep-learning approach.

### 1.2. Major Contributions

Here is a list of this study paper’s main contributions:A fusion-based contrast enhancement technique was proposed for lesion contrast enhancement of the original images.We used a deep learning model called ResNet50 that had already been trained to perform the fine-tuning. After that, deep transfer learning with set hyper-parameters was used to train both the original and enhanced images.After that, a fusing method known as serial mid-value feature fusion was proposed.Flower pollination was proposed, using a controlled regula falsi-based best features selection method before using machine learning classifiers to classify the results.

## 2. Materials and Methods

This section presents a proposed breast cancer classification method based on mammogram images. Figure 1 depicts the proposed framework. The initial mammography images of the datasets are first augmented, and the augmented datasets are then contrasted. The refined Resnet50 deep network was then fed the original and improved images, which were then taught using deep transfer learning. The trained models are then used to determine the characteristics of the global average pool layer. Following feature extraction, the proposed technique is used for fusion. In the following step, we created a feature selection method to select the best features from the fused vector. Finally, machine learning classifiers are used to categorize the best features. Each stage is explained in detail below.

### 2.1. Dataset Collection

In this work, we consider three mammography image datasets: CBIS-DDSM [43], INbreast [44], and MIAS [45]. There are two classes in the CBIS-DDSM dataset, malignant and benign. The second INbreast dataset uses the classifications benign and malignant. The MIAS collection uses the classifications benign, malignant, and normal. Table 1 provides a short description of the images in these datasets. We used data augmentation to carry out all three procedures shown in Figure 2 because these datasets were unbalanced. First, we turned them 90 degrees, and then flipped them to the left and right.

### 2.2. Dataset Augmentation

Recently, several researchers performed data augmentation to improve the learning capability of deep learning methods [46]. For the deep learning model, the data sets currently available in the medical industry are from low-resource areas, but a significant amount of training data are needed [47]. Therefore, a data augmentation process must be used to increase the diversity of the initial datasets. Three datasets are used for validation in this study, as mentioned above. There are 1134 average-sized images in the collection of 500 × 500 pixels in the first CBIS-DDSM dataset. There are two categories in this dataset: Figure 3 shows the number of benign (557) and malignant (637) lesions. The second dataset of breast images contains 146 images with an average resolution of 500 × 500 pixels. This dataset consists of two categories: benign (76) and malignant (70), as shown in Figure 3.

In the third dataset of MIAS, there are 300 images with average sizes from the collection of 500 × 500 Pixels. There are three categories collectively in this dataset: benign, malignant, and normal (209), as seen in Figure 3. We split the entire dataset into training and testing groups (50:50). After training the images for each class, the dataset is inadequate to train the deep learning model. Therefore, three operations—horizontal flip, vertical flip, and rotation 90°—are performed on the original mammography images to increase the dataset diversity. Multiple iterations of these procedures are performed until each class of images contains at least 6000 for CBIS-DDSM and 4000 for INbreast and MIAS. After the augmentation method, the number of images in CBIS-DDSM is 12,000, 8000 in the INbreast dataset, and 12,000 in the MIAS dataset.

**Figure 3 diagnostics-13-01618-f003:**
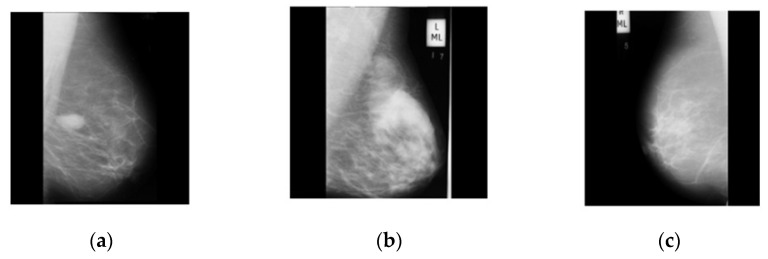
Illustration of mammography images datasets: (**a**) benign class data sample; (**b**) malignant class data sample; (**c**) normal class data sample.

### 2.3. Contrast Enhancement

Enhancing an input image is crucial for improving the original image’s quality. Recently, a lot of technology devices have been introduced for improving image acquisition, but they still they pose several challenges. The challenges include those specific to medical imaging, especially mammography, and are contrast imperfections, chromatic anomalies, and noise. In this proposed technique, our main goal is to improve the infected region and enhance the overall pixels of the image that are more pleasant to the human eye [48]. The objective of this study is to visualize datasets with breast cancer images rather than healthy ones. We created a hybrid approach that relies on combining various filtering results because the breast cancer mammography images’ poor quality and low contrast required us to do so. Considering we have an input image denoted by $\phi \left(x,y\right)$ of size M×N×3, let us have a grayscale image of size M×N denoted by $\tilde{\phi }\left(x,y\right)$. In the first step, the probability density function is applied to the grayscale image as follows:(1)$$p({\tilde{\phi }}_{k})=\frac{h_{k}}{h},k=0,1,2,3,\ldots ,\;L-1$$
where $h_{k}$ denotes the number of pixels having an intensity ${\tilde{\phi }}_{k}$ and h represents the total number of images Δ. In the next step, CDF is computed as follows:(2)$$c({\tilde{\phi }}_{k})\sum_{i=0}^{k}p({\tilde{\phi }}_{k})$$Using the CDF, the transformation function is defined as follows:(3)$$F({\tilde{\phi }}_{k})={\tilde{\phi }}_{0}+({\tilde{\phi }}_{L-1}-{\tilde{\phi }}_{0})*c\left({\tilde{\phi }}_{k}\right)$$
(4)$$\psi =\left\{F(\tilde{\phi }\left(x,y\right)/\forall \;\tilde{\phi }\left(x,y\right))\in \tilde{\phi }\right\}$$After that, the spatial domain transformation is applied on $F({\tilde{\phi }}_{k})$.
(5)$$G(x,y)=\tau \left[F(x,y)\right]$$
(6)$$G(x,y)=Q(x,y)\left[F(x,y)-c\times \mu (x,y)\right]+\mu {\left(x,y\right)}^{a}$$Here, G(x,y) is a transformed image and Q(x,y) is a contrast stretching function that is mathematically defined as follows:(7)$$Q(x,y)=\frac{K\times gm}{\sigma (x,y)+\beta }$$
(8)$$gm=\frac{1}{M\times N}\sum_{x=0}^{M-1}\sum_{y=0}^{N-1}F\left(x,y\right)$$
where gm denotes the global mean value, σ denotes the standard deviation, k and β denote the constant parameters and values manually assigned. Hence, the final transformation function is defined as follows:(9)$$G\left(x,y\right)=\frac{K\times gm}{\sigma \left(x,y\right)+\beta }\left[F\left(x,y\right)-c\times \mu \left(x,y\right)\right]+\mu {\left(x,y\right)}^{a}$$The visual output of the resultant enhanced image $G\left(x,y\right)$ is illustrated in Figure 4.

### 2.4. ResNet50 Deep Learning Features

Deep learning networks acknowledge images as inputs and extract features from them that are later fed to the classifier for classification. This process of feature extraction and classification is conducted automatically [49]. Deep learning methods lower classification error rates to 5%, comparable to human mistakes [50].

ResNet50 [51] is the most frequently utilized subclass of convolution neural networks for picture classification. Figure 5 depicts the ResNet50 design. In this applicable ResNet50 deep learning model, the training was carried out with various weights. A broader network can benefit from extensive training, improving accuracy and outcomes. The ResNet50 network consists of several deep layers with a max-pool layer kernel size of 3 × 3 and a small, related field of 7 × 7 in the input layer.

In this research, we improved this deep algorithm for breast cancer classification. We eliminated the first three layers and introduced three new ones to accomplish this. Later, we clarified the chosen dataset for breast cancer. The training process was conducted using deep transfer learning. Additionally, we did not freeze any single layer during the fine-tuning process.

*Deep Transfer Learning*: A pre-trained model is reused using the machine learning method known as transfer learning for another task [47]. The sampling efficiency could be greatly increased by transferring or reusing data from previously taught tasks for the newly learned activities [52]. TL is used in this case to extract deep features. To perform this, a pre-trained model is initially tweaked before being trained via TL [53]. The process of TL is defined as follows:

The two parameters that comprise the domain $D=\left\{2,\;\right.\left.p\left(z\right)\right\}$ are the feature space z and the distributions of the marginal probabilities $f\left(z\right),$ where $z=\left\{z1,z2,\ldots .\left.zn\right\}\right.\in z$. If two domains are distinct, they either have different marginal probabilities $\left(p\left(zp\right)\neq p\left(zq\right)\right)$ or feature space $\left(zp\neq zq\right)$. Given a specific domain, there are two parts to this task t$\;\left\{\left.s,g\left(.\right)\right\}\right.$: the label space s, and a prediction function *g* (.); this is not immediately apparent but can be obtained using training data $\left\{\;(kj_{,}u_{j}\right.\left.j\left\{1\right.\left.,2,3,4\ldots \ldots .N\right\},where\;k_{j\;and}u_{j\;s}\right\}$; thus, a probabilistic approach *c*$\left(k_{j}\right)$ can be written as *p*$\left(u_{j}|k_{j}\right)$, and so we rewrite this function v as V = $\left\{s,p\left(s|z\right)\right\}$. When two functions are different from each other, their label space is also different $\left(s_{p}\neq s_{q}\right)$ or produces different distributions with conditional probability $\left(p\left(s_{p}|z_{p}\right)\neq p\left(s_{q}|z_{q}\right)\right)$.

Figure 6 gives a detailed overview of the visual transfer learning method. The modified deep model receives the original model’s (source domain) information (target domain) to train this updated model. The following hyperparameters are used in the model: the learning rate is 0.001, the mini-batch size is 16, and there are 200 epochs when using the stochastic gradient descent learning method. The features are extracted from the updated deep model’s Global Average Pooling (GAP) layer. Two feature vectors have been returned in an output that is finally fused using a serial-based mid-value approach.

### 2.5. Novelty 2: Serial Based Mid Value Fusion

Improving an object’s information through the fusion of data from two or more sources is known as information fusion. A serial-based feature is used when putting information from different sources together into a single matrix; the feature fusion method is simple to use while maintaining all features. However, this process added all relevant and irrelevant features. Adding all features increases the computational time and results in maximum chances of a high error rate for the classification purpose. Therefore, in this work, a new mid-value-based function is developed and finds a middle value used for the fusion of final features.

Consider that we have two feature vectors of original and enhanced datasets denoted by f1 and f2. The dimensions of the extracted feature vectors are N×2048 and N×2048, respectively. The serially fused vector dimension will be N×4098 based on the following equation:(10)$$kf_{q\left(v\right)}={\left[\begin{array}{c} f1\\ f2 \end{array}\right]}_{N\times j1+w\times j2}$$This process combined all of the extracted features. Using this fused vector, we computed the mid-value using the following formula:(11)$$Fnc=MidV\left(kf_{q\left(v\right)}\right)$$
(12)$$MidV=\frac{lw+hw}{2}$$The returned middle value is passed to the threshold function for finding the final fused vector.
(13)$$Thers=\left\{\begin{array}{c} Fus\left(f\right)\;for\;kf_{q\left(v\right)}\left(j\right)\geq Fnc\\ Ignore\;Elsewhere \end{array}\right.$$
where $Fus\left(f\right)$ denotes the fused feature vector of dimension N×2772. Further enhancement of this outcome vector was carried out with the flower pollination-controlled regula falsi method.

### 2.6. Feature Optimization

Feature selection involves selecting the best features to improve accuracy and reduce computation time in computer vision. The size of the solution space grows exponentially as the number of features in the dataset increases. According to the theory, using fewer features improves classifier performance. It speeds up classification, resulting in accuracy rates equal to or higher than those obtained when all features are used. Based on the flower pollination-controlled regula falsi, this study developed a feature selection technique (FPcRF). Laws and floral reproduction inspire the flower optimization method. The FPA [54] is used for optimization, feature selection, and other optimization techniques in order to minimize its search space. The final selection requires a fitness function, and in this work, we used Fine-KNN. The initial representation of the flower consistency is
(14)$$\;y_{i^{j+1}}=yi^{j}+\vartheta L\left(\lambda \right)\left(yi^{j}-B\right)\;$$
where $yi^{j}$ is a pollen i or solution vector $y^{i}$ and iteration j and B is the best solution. The symbol ϑ is a given factor of scaling. The Levy Flight Distribution is defined as follows:(15)$$L\sim \lambda \frac{\Gamma \left(\lambda \right)\sin\left(\frac{\Pi }{2}\right)\;/}{\Pi }\frac{1}{s^{1+\lambda }},\;\left(s_{0}>>s_{0}>0\right)$$
where $\Gamma \left(\lambda \right)$ is also due to the accuracy of this distribution and the gamma function for significant steps *s*>0; the local fertilization is shown as follows:(16)$$yi^{j+1}=yi^{j}+µ\left(yz^{j}-yk^{j}\right)\;$$
where $yi^{j}$ and $yk^{j}$ are pollens of different flowers of the same plant.

False Regulation Formula: assume that the number of *y*ij+1 is denoted by r1, the greatest number is represented by r2, and then the root value’s approximation is calculated as follows:(17)V1=r1fr2−r2fr1fr2−fr1
(18)Vn=rn−1frn−rnfrn−1frn−frn−1
where Vn is the final approximate root value of selected vector y ij+1. Based on Vn, the final selection is performed and passed to the fitness function again to check the performance. The selected features are finally classified using machine learning classifiers for the final classification accuracy. Moreover, a detailed algorithm is presented in Algorithm 1.
**Algorithm 1** Proposed Breast Cancer Classification AlgorithmInput: Original Image ϕx,yOutput: Labeled Image ϕ^x,yStep 1: Dataset AugmentationStep 2: Contrast Enhanced using Equations (1)–(9)    -  The resultant Image is denoted by Gx,yStep 3: Trained Deep Learning ModelStep 4: Deep Features Extraction from Original and Enhanced DatasetsStep 5: Features Fusion using Equations (10)–(13)    -  Thers=Fusf    for   kfqvj≥FncIgnore                      ElsewhereStep 6: Best Feature Selection    -  Objective function: minimum and maximum fx,x=x1,x2…̤xd
    -  Initialization: A population of n flower with a random solution    -  Find the best solution: g∗ initial population    -  Probability: p∈0,1
**While** j<max_iterationfor i = 1: nif rand <p,Draw step levy distribution is given asGlobal pollination via yij+1=yij+ϑLλyij−B
ElseDraw Q from a uniform distribution 0,1
Local pollination via L~ λ ΓλsinΠ2 /Π 1s1+λ, s0>>s0>0end ifEvaluate new solutionendfor best solutionFind best root using Equations (17)–(18)Final Best Solution**end while**

## 3. Experimental Results and Discussion

### 3.1. Datasets and Experiments

Extensive experiments have been performed for the analysis of the proposed framework. Graphical representations and tabular data are used to show the findings. For the experimental procedure, three datasets have been used, and details are provided in Section 3.1. Furthermore, several experiments have been performed, such as proposed fusion and feature selection. In addition, the intermediate results are also provided to show the efficiency of the proposed framework.

### 3.2. Experimental Setup

The suggested framework’s training-to-testing ratio is set at 50/50. Several hyperparameters have been used during the design and experimental process, including a learning rate of 0.0002, epochs of 100, mini-batch size of 32, momentum value of 0.7223, and stochastic gradient descent (SGD) as an optimizer. During the testing procedure, a 10-fold cross-validation was used for each experiment. Various metrics were used to evaluate the success of the various classifiers, including F1-Score, FPR, Kappa, MCC, Accuracy, and Time as examples of statistical measures. A desktop computer was used, by one individual, with 16 GB of RAM and an 8 GB graphics card running MATLAB2022a, which simulated the full experimental procedure.

### 3.3. CBIS-DDSM Breast Cancer Dataset Results

The breast cancer information from the CBIS-DDSM dataset that was recommended for merging is displayed in Table 2. The cubic SVM (CSVM) had a maximum accuracy of 93.8%. The other calculated CSVM performance metrics had a precision rate of 93.09%, F1-score of 93.89%, FPR of 0.07, Kappa and MCC of 86.87%, and Sensitivity rates of 94.70% and 87.68%, respectively. These results for the remaining classifications demonstrate that CSVM surpasses them all. Figure 7 also displays the CSVM confusion matrix, which can be used to confirm the claimed performance metrics. The computation times for each classifier are also given; the MNN classifier recorded the lowest computation time of 249.39 s (seconds). Figure 8 shows the computational time mentioned previously.

Based on the suggested FPcRF, Table 3 presents the optimal feature selection results from the CBIS-DDSM. The CSVM classifier has a maximum accuracy of 93.3%. Additional metrics include a precision rate of 92.26%, sensitivity rate of 94.57%, F1-score of 93.40%, FPR of 0.07, Kappa of 86.63%, and MCC of 86.66%, supporting the proposed findings. Figure 9 depicts the CSVM’s confusion matrix, which can be used to verify computed performance measures. The assessment procedure also tracks how long each classification’s computations take. From lowest to highest, the spread is 82.532 (s) to 503.23 (s). Figure 8 visually compares the stated computational times for each classifier for both trials. The computational time is drastically decreased when the suggested FPcRF selection method is used, as is abundantly clear from this figure. Additionally, overall accuracy is improved after employing the recommended feature selection approach.

### 3.4. INbreast Breast Cancer Dataset Results

Table 4 displays the outcomes of the INbreast breast cancer dataset’s suggested fusion. The narrow neural network (NNN) classification achieved the highest accuracy of 99.5%, with values of 99.35%, 99.65%, 99.50%, and 99.00% for the precision rate, sensitivity, F1-score, Kappa, and MCC, respectively. These numbers are also calculated for the remaining classifiers, and the NNN’s best performance is discovered. Figure 9 shows a confusion matrix for NNN. The malignant class accurate prediction rate is shown in this figure to be above 99%. Each classifier’s computational time is mentioned, and the min-max range is 100.94–289.57 (s). Overall, the NNN produced better outcomes but took longer, which is a disadvantage.

We suggested a feature selection technique to address this flaw, and the results are shown in Table 5. This table shows that the best accuracy, 99.6%, which was attained using a bi-layered NN. A 99.45% accuracy rate, a 99.65% sensitivity rate, an F1-score of 99.55%, an FPR of 0.05, a Kappa of 99.10%, and an MCC of 99.10% are all the values reported. These metrics are calculated using various algorithms, including SVM, neural networks, and fine-KNN. Figure 10 and Figure 11 depicts the confusion matrices that displays the correct rate for each class, such as 99.45% for the malignant class.

It should be observed that the accuracy for the maximum classifier is consistent with Table 4, and for some classifiers it is improved. Each classifier’s computational duration, which can range from 127 s to 39.88 s, is also noted. Figure 12 compares proposed fusion and selection techniques and demonstrates how much less time is needed when using FPcRF selection.

**Table 4 diagnostics-13-01618-t004:** Classification results of proposed feature fusion for INbreast dataset.

Classifier	Precision	Sensitivity	F1-Score	FPR	Kappa	MCC	Accuracy	Time (s)
CSVM	99.15	99.70	99.43	0.08	98.85	98.85	99.4	123.89
LSVM	98.37	99.60	98.98	0.01	97.95	97.96	99.0	121.42
QSVM	99.20	99.55	99.38	0.08	98.75	98.75	99.4	116.32
MGSVM	98.71	99.45	99.08	0.01	98.15	98.15	99.1	179.52
MNN	99.25	99.70	99.48	0.07	98.95	98.95	99.5	108.77
WNN	99.25	99.60	99.43	0.07	98.85	98.85	99.4	150.39
FKNN	98.70	99.00	98.85	0.01	97.70	97.70	98.9	289.57
**NNN**	**99.35**	**99.65**	**99.50**	**0.06**	**99.00**	**99.00**	**99.5**	**101.52**
BNN	98.45	99.40	99.42	0.05	98.85	98.85	99.4	103.49
TNN	99.35	99.40	99.38	0.06	98.75	98.75	99.4	100.94

**Table 5 diagnostics-13-01618-t005:** Classification results for INbreast dataset using proposed feature selection technique.

Classifier	Precision	Sensitivity	F1-Score	FPR	Kappa	MCC	Accuracy	Time (s)
CSVM	99.35	99.65	99.50	0.06	99.0	99.0	99.5	49.65
LSVM	98.32	99.60	98.96	0.01	97.90	97.91	99.0	54.65
QSVM	99.30	99.65	99.48	0.07	98.95	98.95	99.5	46.45
MGSVM	98.66	99.55	99.10	0.01	98.20	98.20	99.1	78.90
MNN	99.25	99.50	99.38	0.07	98.75	98.75	99.4	42.14
WNN	99.35	99.45	99.40	0.06	98.80	98.80	99.4	63.26
FKNN	98.90	99.20	99.05	0.01	98.10	98.10	99.1	127.00
NNN	99.55	99.50	99.52	0.04	99.05	99.05	99.5	40.01
**BNN**	**99.45**	**99.65**	**99.55**	**0.05**	**99.10**	**99.10**	**99.6**	**39.88**
TNN	99.35	99.45	99.40	0.06	98.80	98.80	99.4	40.75

**Figure 10 diagnostics-13-01618-f010:**
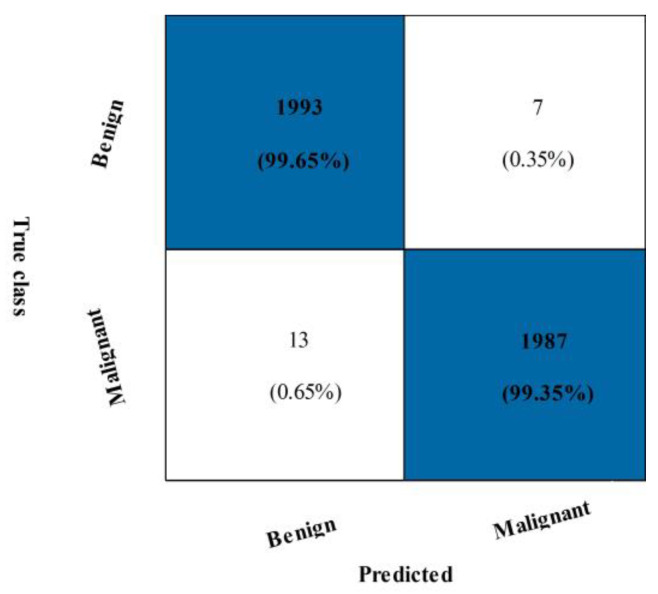
Confusion matrix of NNN classifier for INbreast dataset using proposed fusion technique.

**Figure 11 diagnostics-13-01618-f011:**
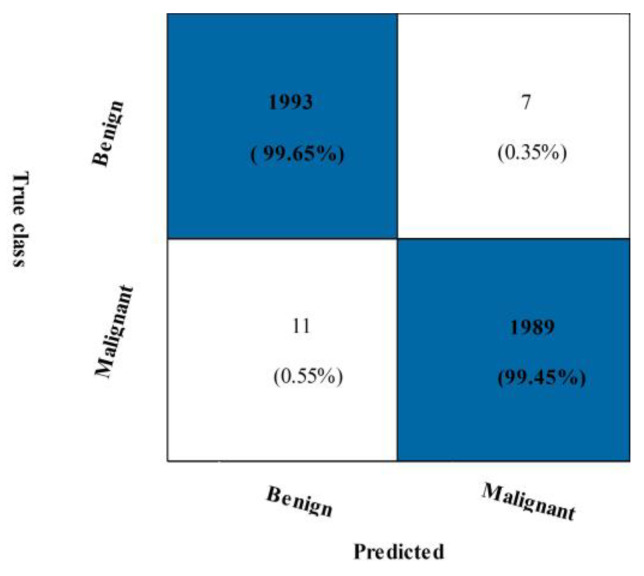
Confusion matrix of BNN classifier using proposed feature selection technique on CBIS-DDSM dataset.

**Figure 12 diagnostics-13-01618-f012:**
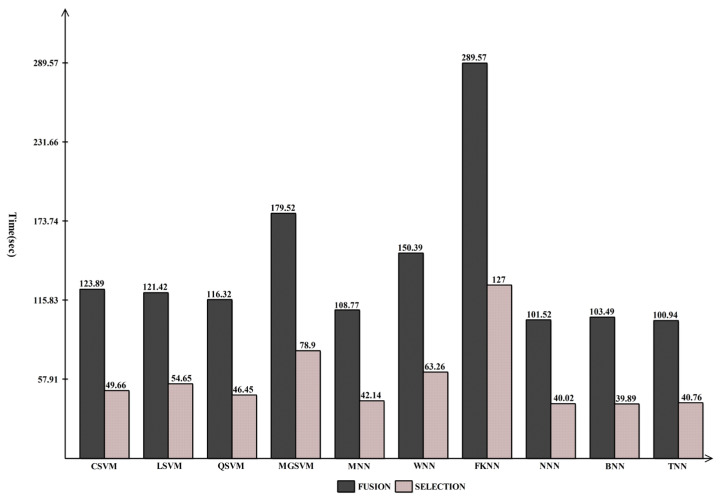
Comparison of proposed fusion and selection steps in terms of computational time for INbreast dataset.

### 3.5. Mias Dataset Results

Table 6 shows the outcomes of the suggested fusion technique on the MIAS dataset. The NNN classifier achieved maximum accuracy at 99.8%. Several additional performance metrics have also been calculated, including the precision rate, sensitivity rate, F1-score, Kappa value, and MCC value, which are all 99.82, 99.60, 99.82, and 99.73%, respectively. Figure 13, which displays the computed values as a confusion matrix, can be used to validate them. The benign and malignant classes in this figure had 100% and 99.7% accurate prediction rates. Each classifier’s computational duration is also listed, taking a maximum of 556.93 s and a minimum of 106.68 s.

Overall, the CSVM produced better outcomes, but this experiment’s disadvantage was that it took longer. To address this issue, a feature selection method is suggested. The outcomes are shown in Table 7. The CSVM yields the highest accuracy of 99.8% for this experiment. In particular, the Kappa is 99.50%, the MCC is 99.67%, the sensitivity is 99.78%, the F1-score is 99.78%, and the precision rate is 99.78%. SVM, neural networks, and fine KNN are just a few algorithms used to calculate all these metrics. Figure 14 shows a confusion matrix with the accurate prediction rates for each class, respectively, 99.7% and 100% for benign and malignant classes. It should be noted that the accuracy is consistent with Table 6, while the computational time is considerably shorter. Finally, the proposed fusion and selection techniques are compared in Figure 15, which illustrates a substantial reduction in computational time.

### 3.6. Comparison with Other State-of-the-Art Techniques

The proposed strategy is contrasted with a few contemporary techniques in Table 8. In [23], the authors proposed a multi-fractal and fusion approach and obtained an accuracy of 98% using three mammography breast cancer datasets, including CBIS-DDSM, INbreast, and MIAS. The precision for CBIS-DDSM, INbreast, and MIAS was 98.4% using the CNN-based model [46]. In this work, the authors used capsule network based architecture. The CNN-based architecture and fusion method were used in [47]. They obtained 96% accuracy, later improved by [48] using modified deep learning architecture, and achieved an average of 99.0% accuracy. The proposed framework obtained an accuracy of 93% on the CBIS-DDSM dataset, 99.5% on the INbreast dataset, and 99.8% on the MIAS dataset. These values show the improvement in accuracy compared to the existing techniques.

### 3.7. Confidence Interval-Based Analysis

A detailed confidence interval-based analysis is conducted in this section for all three datasets. The confidence interval (CI) is computed based on the Kappa measure. For the CBIS-DDSM dataset, the minimum and maximum noted values are 77.13% and 86.63%. Using these values, the obtained margin of error (MoE) is illustrated in Figure 16. In this figure, it is noted that at a confidence level of 68.3%, σx̄, the margin of error is 82.88 ± 4.066 (±4.91%). For a confidence level of 95%, 1.960σx̄, the margin of error is 82.88 ± 7.969 (±9.62%). This shows that the error dropped to 5% when a confidence level changed. Similarly, the MoE is computed for the rest of both datasets (Inbreast and MIAS), and the values are illustrated in Figure 17. For the Inbreast dataset, the MoE is 98.5 ± 0.832 (±0.84%) at a confidence level of 95%, 1.960σx̄. For the MIAS dataset, the MoE is 97.975 ± 2.114 (±2.16%) at a confidence level of 95%, 1.960σx̄. This shows that the performance of the proposed method is sufficient.

### 3.8. Visual Facts

In the end, the visualization is performed for the proposed deep learning framework. The Grad-CAM visualization is employed for the gradient visualization of the proposed deep framework, as shown in Figure 18. In this figure, a heatmap is applied to the important region based on the grad-CAM. The brown color shows the most important region that reveals this framework’s performance. In addition, the labeled prediction results are illustrated in Figure 19. Based on this figure, we can see that the proposed framework correctly predicts the benign and malignant classes. Finally, we also added a brief analysis for the selection of training and testing ratios. As shown in Figure 20, accuracy is consistent and only a minor change occurred for our selected ratio of 50:50. However, it is noted that the accuracy is going to decrease when the ratio is 40:60 or 30:70.

## 4. Conclusions

Breast cancer is the leading cause of cancer death in women worldwide. The early detection of malignant and benign cancers can help patients receive timely treatment. This study proposed a new framework based on fusion-based contrast enhancement using deep learning features from the average pool layer. It proposed feature selection for breast cancer classification based on mammography images. While some of the original points were lost during the fusion process as a result of the contrast enhancement phase, the performance of the fine-tuned deep learning model outperformed that of the original images. The fusion of deep features reduced the proposed framework’s effectiveness while improving accuracy and overall computational time. As a result, we proposed a feature selection method that improved precision while reducing computational time. Furthermore, the proposed framework improves accuracy for all three selected datasets by 93, 99.5%, and 99.8%, respectively. This work’s main limitations are as follows: i) the addition of a contrast enhancement technique improves image quality but increases computational time; ii) two consecutive constant values are not always obtained in the selection step, and the entire algorithm is executed for all initialized iterations. This takes longer and can result in the loss of important features.

### Future Scope

Propose a new method for improving the contrast of the infected region pixels, which will improve the visibility of the lesion region and help with the correct segmentation.Propose a segmentation technique using deep learning and a saliency map for tumor detection. The residual block will be added in the deep learning model, which can aid in better learning for the detection process.Develop a fusion-based deep learning architecture with Bayesian optimization-based hyperparameters initialization.Propose a new feature selection technique that will stop the iteration in a maximum of two constant cost values.

## Figures and Tables

**Figure 1 diagnostics-13-01618-f001:**
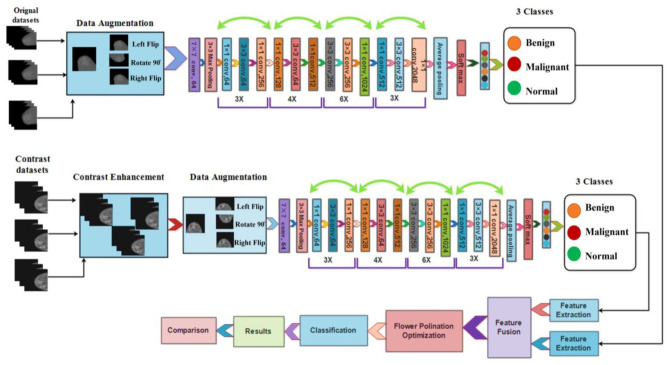
The proposed framework for classifying breast cancer from mammography images.

**Figure 2 diagnostics-13-01618-f002:**
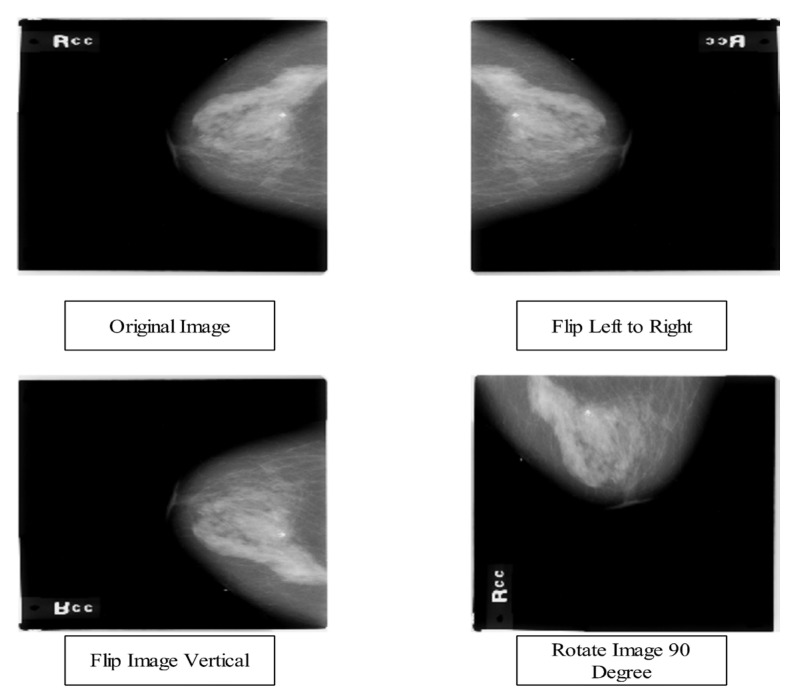
Visual samples of data augmentation on the selected dataset. * Started from the original image and obtained three new images after employing operations such as flip left to right, flip vertical, and rotate 90.

**Figure 4 diagnostics-13-01618-f004:**
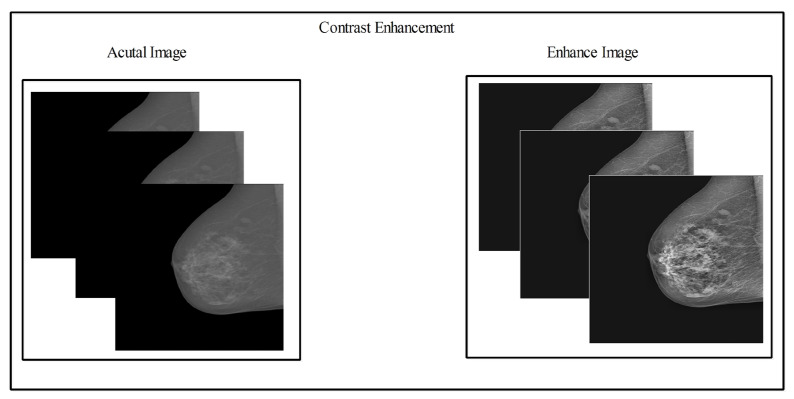
For improved visualization of the images, contrast-enhancing methods were used. **Left** side: Original images before any image enhancement technique; **Right** Side: improved images after applying the proposed fused contrast enhancement method.

**Figure 5 diagnostics-13-01618-f005:**
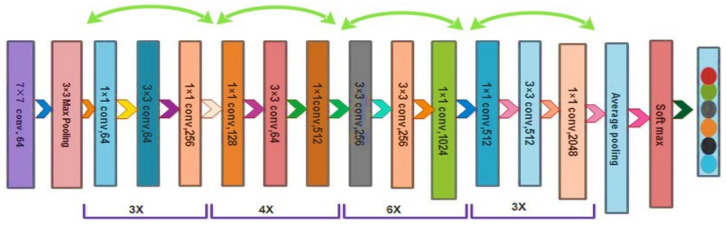
The architecture of ResNet50 model.

**Figure 6 diagnostics-13-01618-f006:**
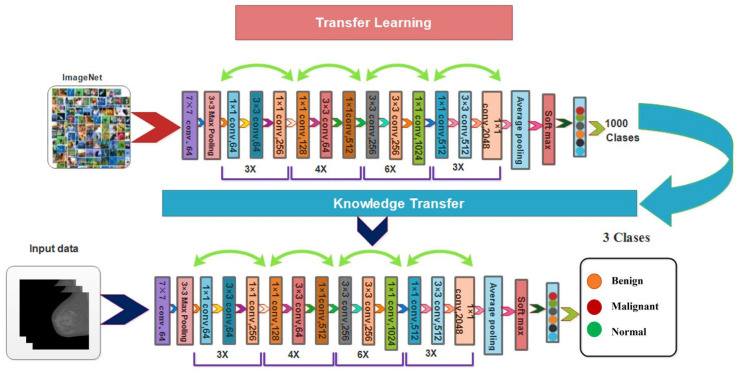
Transfer learning-based training of modified model and feature extraction.

**Figure 7 diagnostics-13-01618-f007:**
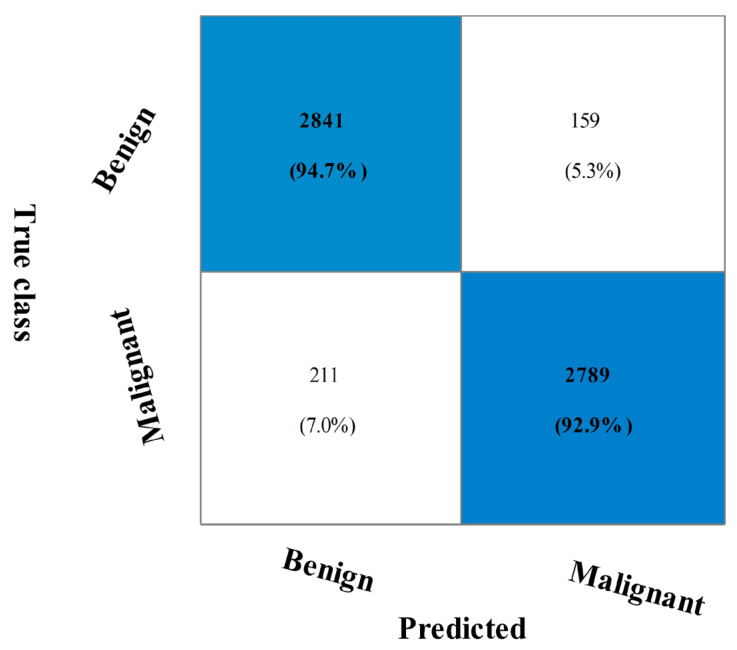
Confusion matrix of CSVM on CBIS-DDSM dataset using proposed fusion technique.

**Figure 8 diagnostics-13-01618-f008:**
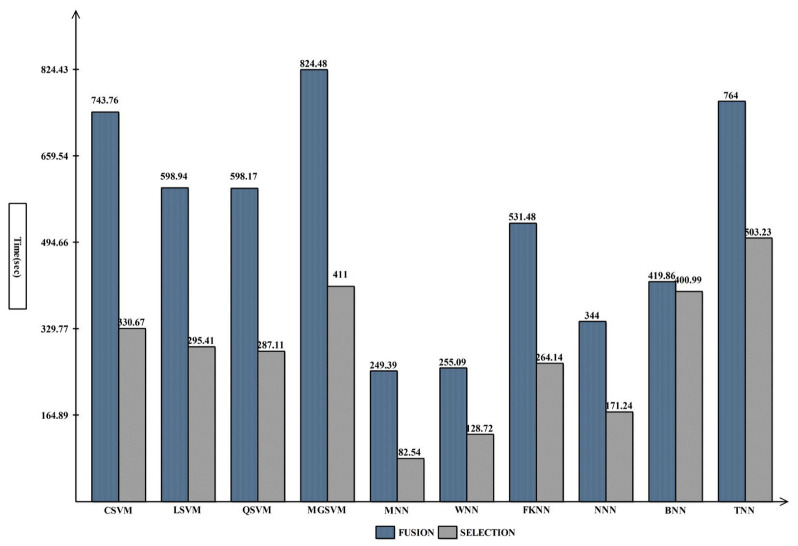
Comparison of proposed fusion and selection steps in terms of computational time.

**Figure 9 diagnostics-13-01618-f009:**
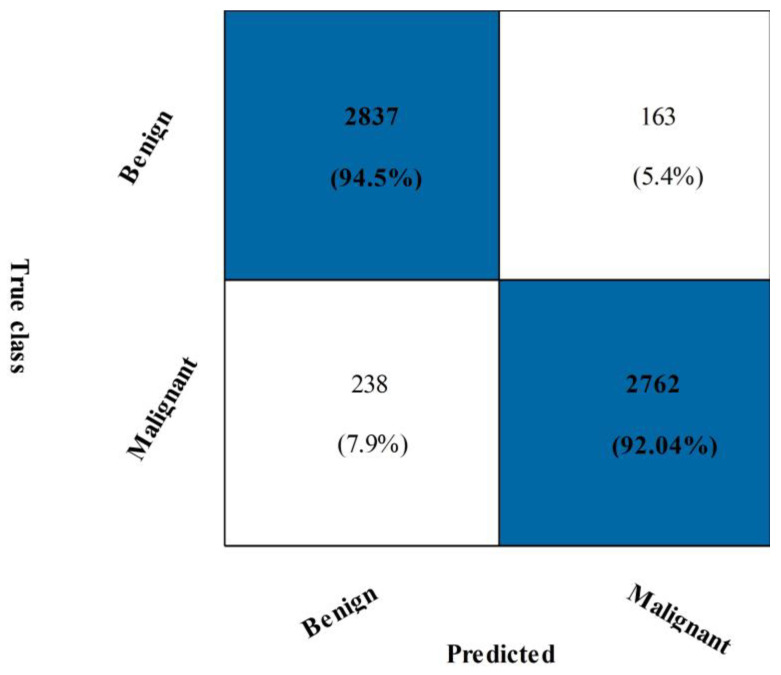
Confusion matrix of CSVM on CBIS-DDSM dataset using proposed feature selection technique.

**Figure 13 diagnostics-13-01618-f013:**
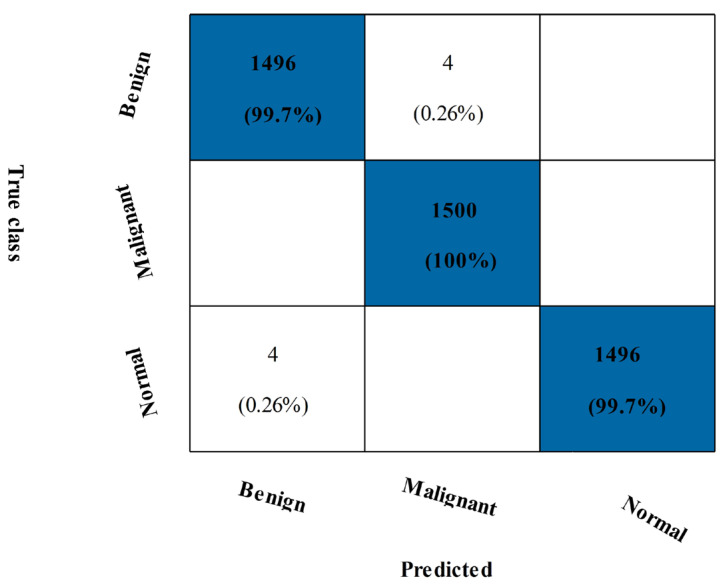
Confusion matrix of MIAS dataset of CSVM Classifier after fusion.

**Figure 14 diagnostics-13-01618-f014:**
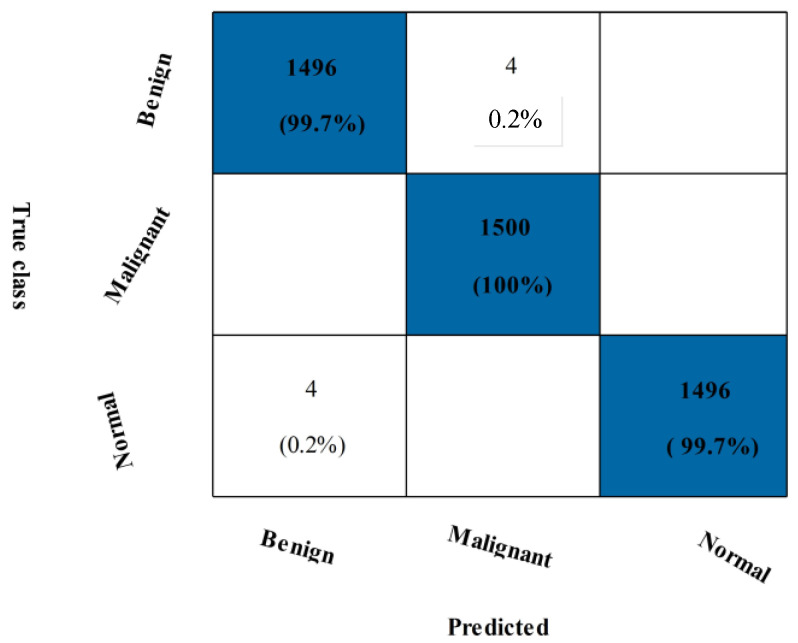
Confusion matrix of MIAS dataset for CSVM Classifier after proposed feature selection technique.

**Figure 15 diagnostics-13-01618-f015:**
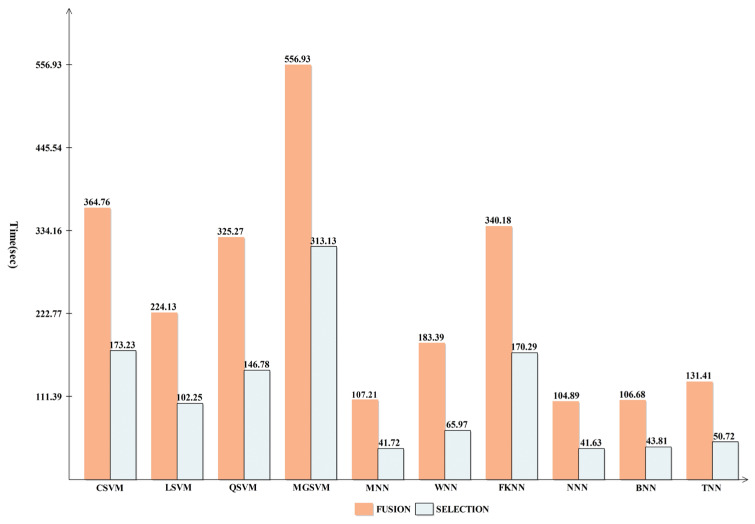
Time-based comparison among proposed fusion and selection techniques for MIAS dataset.

**Figure 16 diagnostics-13-01618-f016:**
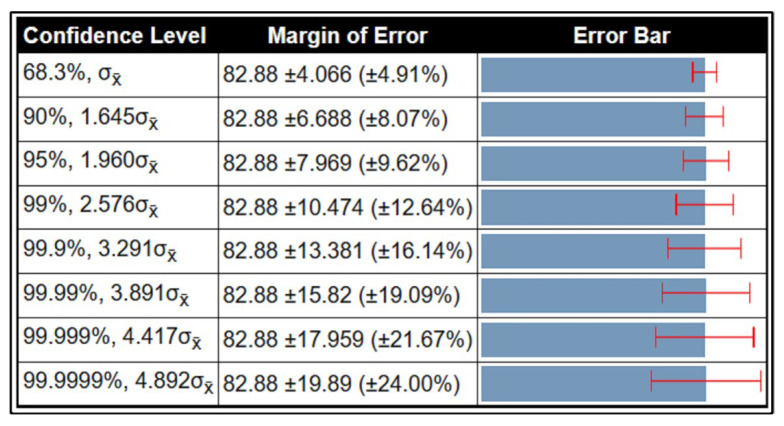
CI-based analysis of proposed framework on CBIS-DDSM dataset using Kappa measure.

**Figure 17 diagnostics-13-01618-f017:**
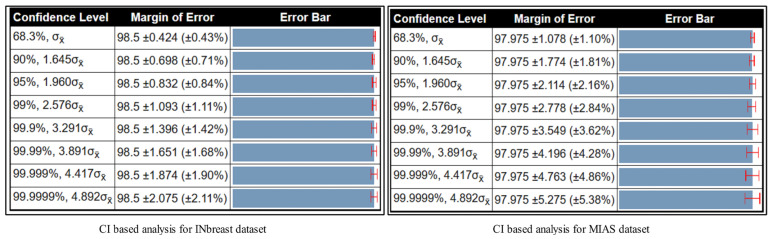
CI-based analysis of proposed framework on INbreast and MIAS datasets.

**Figure 18 diagnostics-13-01618-f018:**
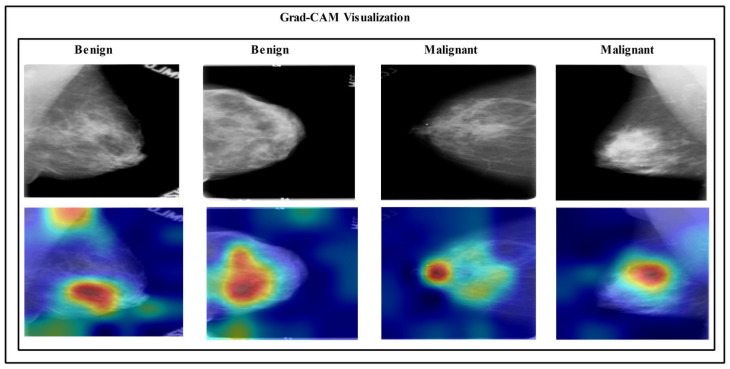
Grad-CAM-based visualization of proposed deep learning framework.

**Figure 19 diagnostics-13-01618-f019:**
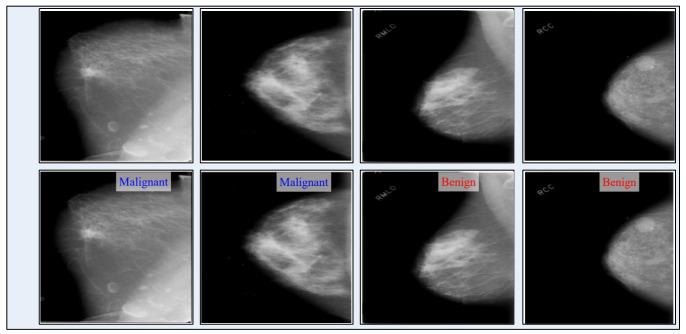
Label-based visual prediction results of the proposed framework.

**Figure 20 diagnostics-13-01618-f020:**
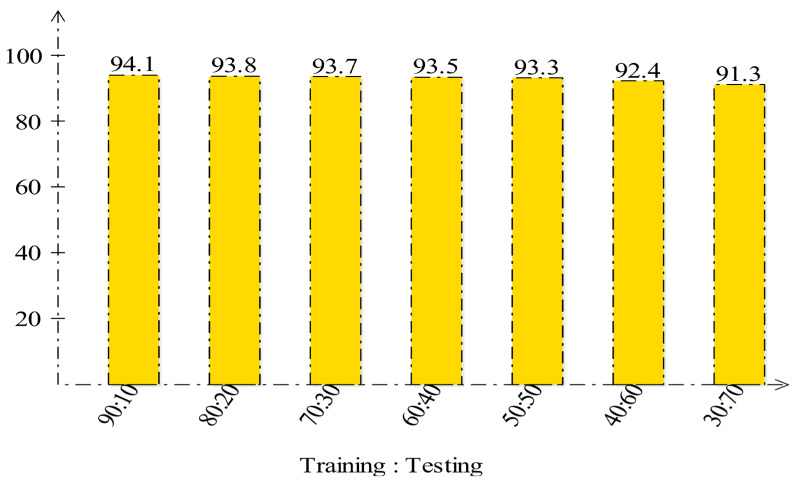
Accuracy analysis for CBIS-DDSM dataset after employing several training and testing ratios.

**Table 1 diagnostics-13-01618-t001:** Brief description of selected datasets.

Dataset Name	Classes	Images	Augmented	Training/Testing
CBIS-DDSM	Benign	557	6000	3000/3000
	Malignant	637	6000	3000/3000
INbreast	Benign	76	4000	2000/2000
	Malignant	70	4000	2000/2000
MIAS	Benign	52	4000	2000/2000
	Malignant	39	4000	2000/2000
	Normal	209	4000	2000/2000

**Table 2 diagnostics-13-01618-t002:** Classification results of proposed feature fusion CBIS-DDSM dataset.

Classifier	Precision	Sensitivity	F1-Score	FPR	Kappa	MCC	Accuracy	Time (s)
**CSVM**	**93.09**	**94.70**	**93.89**	**0.07**	**87.67**	**87.68**	**93.8**	**743.76**
LSVM	89.83	90.67	90.25	0.10	80.40	80.40	90.2	598.94
QSVM	92.03	93.20	92.61	0.08	85.13	85.14	92.6	598.17
MGSVM	91.55	92.47	92.01	0.08	83.93	83.94	92.0	824.48
MNN	90.79	92.03	91.41	0.09	82.70	82.71	91.3	249.39
WNN	91.45	92.00	91.72	0.08	83.40	83.40	91.7	255.09
FKNN	91.96	93.73	92.84	0.08	85.53	85.55	92.8	531.48
NNN	90.03	90.57	90.30	0.10	80.53	80.53	90.3	344.00
BNN	89.37	89.63	89.50	0.01	78.97	78.97	89.5	419.86
TNN	88.86	89.90	89.38	0.11	78.63	78.64	89.3	764.00

**Table 3 diagnostics-13-01618-t003:** Classification results of proposed FPcRF-based feature selection for CBIS-DDSM dataset.

Classifier	Precision	Sensitivity	F1-Score	FPR	Kappa	MCC	Accuracy	Time (s)
**CSVM**	**92.26**	**94.57**	**93.40**	**0.07**	**86.63**	**86.66**	**93.3**	**330.67**
LSVM	89.04	89.87	89.45	0.11	78.80	78.80	89.4	295.41
QSVM	90.95	92.77	91.85	0.09	85.53	83.55	91.8	287.11
MGSVM	91.03	92.30	91.66	0.091	83.20	83.21	91.6	411.00
MNN	89.99	90.50	90.24	0.10	80.43	80.43	90.2	82.53
WNN	90.47	90.80	90.63	0.09	81.23	81.23	90.6	128.72
FKNN	92.06	93.50	92.77	0.08	85.43	85.44	92.7	264.14
NNN	88.48	89.37	88.92	0.11	77.73	77.74	88.9	171.24
BNN	88.81	89.40	89.10	0.11	78.13	78.14	89.1	400.99
TNN	87.91	89.43	88.66	0.12	77.13	77.14	88.6	503.23

**Table 6 diagnostics-13-01618-t006:** Classification results of proposed features fusion for MIAS dataset.

Classifier	Precision	Sensitivity	F1-Score	FPR	Kappa	MCC	Accuracy	Time (s)
**CSVM**	**99.82**	**99.82**	**99.82**	**0.08**	**99.60**	**99.73**	**99.8**	**364.76**
LSVM	99.49	99.49	99.49	0.02	98.85	99.23	99.5	224.13
QSVM	99.76	99.76	99.76	0.01	99.45	99.63	99.8	325.27
MGSVM	99.82	99.82	99.82	0.08	99.60	99.73	99.8	556.93
MNN	99.56	99.56	99.56	0.02	99.00	99.33	99.6	107.21
WNN	99.67	99.67	99.9	0.01	99.25	99.50	99.7	183.39
FKNN	98.50	98.47	98.46	0.07	96.55	97.72	98.5	340.18
NNN	99.51	99.51	99.51	0.02	98.90	99.27	99.5	104.89
BNN	99.47	99.47	99.47	0.02	98.80	99.20	99.5	106.68
TNN	98.98	98.98	99.98	0.05	97.70	98.47	99.0	131.41

**Table 7 diagnostics-13-01618-t007:** Classification results of proposed feature selection technique for MIAS dataset.

Classifier	Precision	Sensitivity	F1-Score	FPR	Kappa	MCC	Accuracy	Time (s)
**CSVM**	**99.78**	**99.78**	**99.78**	**0.01**	**99.50**	**99.67**	**99.8**	**173.23**
LSVM	99.34	99.33	99.33	0.03	98.50	99.00	99.3	102.25
QSVM	99.69	99.69	99.69	0.01	99.30	99.53	99.7	146.78
MGSVM	99.82	99.82	99.82	0.08	99.60	99.73	99.8	313.13
MNN	99.45	99.44	99.44	0.02	98.75	99.17	99.4	41.72
WNN	99.51	99.51	99.51	0.02	98.90	99.27	99.5	65.96
FKNN	98.47	98.42	98.42	0.07	96.45	97.65	98.3	170.29
NNN	99.36	99.36	99.36	0.03	98.55	99.04	99.4	41.62
BNN	98.96	98.96	98.96	0.05	97.65	98.44	99.0	43.81
TNN	98.69	98.69	99.69	0.06	97.05	98.04	98.7	50.72

**Table 8 diagnostics-13-01618-t008:** Comparison with other state-of-the-art techniques.

Model	Year	Method	F1-Score	Accuracy
[55]	2022	Capsule Neural Network Model	CBIS-DDSM/ INbreast/MIAS	98.4
[56]	2017	All Convolutional Design	CBIS-DDSM/ INbreast/MIAS	96
[57]	2022	CoroNet	CBIS-DDSM/ INbreast/MIAS	99.7
Proposed Method	-	Fine-Tuned ResNet50 Model, Fusion & Flower Pollination Optimization Algorithm	CBIS-DDSM/ INbreast/MIAS	93.3% 99.5% 99.8%

## Data Availability

The datasets used in this work are publicly available.

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
