# Peer review of "BRMI-Net: Deep Learning Features and Flower Pollination-Controlled Regula Falsi-Based Feature Selection Framework for Breast Cancer Recognition in Mammography Images"

_diagnostics, 2023, doi:10.3390/diagnostics13091618_

Round 1

Reviewer 1 Report

Its a interesting paper with use of current stat of the art technology for making novel approach for generate accuracy of biological mapping. Still have question about why the proposed work shows CBIS-DDSM showing 93% accuracy compared with other approach and reported literatures. Author need to explain physical meaning behind it. 

Author Response

Response Sheet is attached. thank you

Reviewer 2 Report

See attached file.

Author Response

Response sheet attached. thank you for your valuable comments. 

Reviewer 3 Report

The authors present their original study on development of deep learning algorithms for breast cancer diagnosis using mammography. 

I do have several comments for the authors:

1. I don't know why the authors separate the related work from the Introduction section. Related work should be a part of the Introduction that should focus on identifying the current problem  in this field and establishing research hypothesis. These two parts should be combined and rewritten.

2. For the methods section, the information of how they distributed the samples for training, verification and testing should be included.

3. The rationale for enhancing the images was not justified.

4. I would suggest the authors to include a separate paragraph to summarize the statistic analysis method.

5. The authors should discuss the limitations of there study.

Author Response

Response sheet is attached. thank you for your valuable comments

Reviewer 4 Report

The authors proposed a feature selection framework by using flower pollination controlled regula falsi-based optimization techniques which were extracted by deep learning model for the classification of breast cancer from Mammography images. The proposed study has many caveats to address including the structure of the article, way all the findings are reported and may not be suitable for publication for the following reasons.

1.      It is indeed common employ to feature extraction and selection when necessary. However, it is not clear as to why we can’t use the Mammography images as it is for the classification in this particular case? What classification accuracy do we anticipate if we use the images in their pure form? As mentioned by the authors, deep learning or machine learning techniques are powerful then why there is a need for feature selection in the first place?

2.      Since the study is related to feature selection to improve the classification, in the “Related work” section, it is advisable to promote the advantages of this study over other feature selection-related studies and mention the drawbacks of other studies which can be overcome by the author’s proposal.

3.      The data augmentation can be conveyed if the original image can also be shown with augmented images. The author must try to convey the information as clearly as possible to the readers.

4.      Also, it is not clear whether the augmentation was performed on the whole dataset or only on the training dataset. If it was performed on the whole dataset, then was it ensured that the splitting was done correctly to avoid data leakage?

5.      Abbreviations used first time in a text should be given by their full name so that the readers can understand the author's message clearly.

6.      Usually, the future scope is nothing but applications of the study or possible modifications in the existing methodology to improve the results or performance. The authors must mention the limitations of any of the proposed methods or correctly convey the future scope.

Author Response

Response sheet attached. Thank you for your valuable comments. 

Round 2

Reviewer 2 Report

Please, see attached file.

Author Response

Dear reviewer, thank you for your valuable comments. We thoroughly revised our manuscript. The response sheet is also attached. thank you. 

Reviewer 3 Report

The authors have addressed most of my previous  comments, and the manuscript has been significantly improved after revising. But I still have one minor suggestion of the structure of the paper for the authors.

The discussion of the significance and contributions of the study should be included in the discussion and conclusion section, instead of in the Introduction section.

Author Response

The authors have addressed most of my previous comments, and the manuscript has been significantly improved after revising. But I still have one minor suggestion of the structure of the paper for the authors.

The discussion of the significance and contributions of the study should be included in the discussion and conclusion section, instead of in the Introduction section

Response: Authors are thankful to honorable reviewer for such a valuable recommendation. As per journal template, we added the major contributions and significance of the techniques under the introduction section; however, we also analyze the significance of major contributions under the results section and conclude under the conclusion section. Please see section 3.7 and 3.8. thank you

Reviewer 4 Report

The authors addressed raised concerns partially. However, the article structure is not suitable according to journal guidelines.

Author Response

Dear reviewer, thank you for your valuable recommendation. We carefully revised the manuscript and followed the structure of this manuscript per the journal authors' guidelines. thank you

Response: Authors are thankful to honorable reviewer for valuable comments. We tried our best to update the manuscript according to recommendations. Moreover, we also followed the journal template.

In the journal template, the following heading should be consider such as: Introduction, Material and Methods, Results and discussion, and conclusion.

In the revised manuscript, we followed the same headings with few other sub-headings are added such as major contributions, major challenges, and future works. Thank you